# Impact of COVID-19 Vaccination on Heart Rate Variability: A Systematic Review

**DOI:** 10.3390/vaccines10122095

**Published:** 2022-12-07

**Authors:** Chan-Young Kwon, Boram Lee

**Affiliations:** 1Department of Oriental Neuropsychiatry, Dong-Eui University College of Korean Medicine, 52-57, Yangjeong-ro, Busanjin-gu, Busan 47227, Republic of Korea; 2KM Science Research Division, Korea Institute of Oriental Medicine, 1672, Yuseong-daero, Yuseong-gu, Daejeon 34054, Republic of Korea

**Keywords:** COVID-19, SARS-CoV-2, vaccination, HRV, RMSSD

## Abstract

Establishing and disseminating evidence-based safety information could potentially facilitate beneficial choices in coronavirus disease (COVID-19) vaccinations. This systematic review investigated the potential impact of COVID-19 vaccinations on human heart rate variability (HRV) parameters through comprehensive searches of four electronic medical databases. Five observational studies reporting HRV parameters of individuals vaccinated against COVID-19 and published up to 29 July 2022 were included in this review. Among them, four studies reported the square root of the mean squared differences of successive NN intervals (RMSSD) as their outcome, and the remaining study reported an HRV-based stress indicator. These studies reported short-term changes and rapid recovery in HRV parameters within up to 3 days after COVID-19 vaccination. Some studies showed that the impact of COVID-19 vaccinations on RMSSD was greater in women than men, and in the younger group than in the older group. The methodological quality of the included studies was not optimal; the review revealed short-term changes in HRV parameters, particularly RMSSD, following COVID-19 vaccination. However, as the included studies did not report important parameters besides RMSSD, the limitation exists that the postvaccination long-term HRV stability was not reported.

## 1. Introduction

The coronavirus disease (COVID-19) pandemic has necessitated the development of vaccines against the severe acute respiratory syndrome coronavirus 2 (SARS-CoV-2), and the Pfizer/BioNTech’s BNT162b2 vaccine is the first COVID-19 vaccine that was approved by the U.S. Food and Drug Administration (FDA) [1]. With the emergence of SARS-CoV-2 variants that pose a challenge to the eradication of COVID-19, efforts to develop more effective COVID-19 vaccines have continued [1]. Furthermore, concerns about vaccine safety are a contributory factor for the high prevalence of COVID-19 vaccination hesitancy (~25%) [2]. Therefore, establishing and disseminating evidence-based safety information on COVID-19 vaccines could potentially facilitate COVID-19 vaccine uptake [3].

A recent review of COVID-19 vaccination-associated adverse events identified common adverse reactions, including localized pain and swelling at the injection-site, fever, headache, myalgia, and chills [4]. Specifically, mRNA vaccines were associated with myocarditis, glomerular diseases, and cutaneous eruptions, whereas adenoviral vector vaccines were associated with cerebral venous thrombosis [4]. Moreover, neurological adverse events constitute an important proportion of COVID-19 vaccination-associated adverse events. An analysis of pharmacovigilance data from the World Health Organization database revealed that, although causality remains poorly established, vaccine administration was likely associated with neurological symptoms, including dizziness, headache, lethargy, migraine, parosmia, and poor sleep quality [5].

Despite the lack of systematic, rigorous reporting, COVID-19 vaccination may be related to autonomic system (ANS) dysfunction, which has been reported in some case reports [6,7,8,9,10,11]. However, it is unclear whether COVID-19 vaccination directly causes autonomic dysfunction, although anecdotal cases are likely to increase public anxiety and, consequently, vaccination hesitancy. The possible associations of other vaccines, including human papillomavirus (HPV) [12] and influenza A [13], with autonomic dysfunction, that were previously reported may be partially explained by the systemic inflammatory response [14], and multisystem inflammatory syndrome (MIS) cases associated with COVID-19 vaccination have been reported recently [15]. Thus, the widespread public health implications of COVID-19 vaccination underscore the public health importance of ascertaining the ANS effects of COVID-19 vaccines.

Heart rate variability (HRV) is an important and objective indicator for evaluating the regulation of autonomic balance [16], and its association with inflammatory conditions has been extensively investigated. HRV has indicated an association between influenza vaccination and ANS dysfunction [14]. Williams et al. analyzed data from 51 clinical studies in a meta-analysis and concluded that HRV-related variables, especially the standard deviation of the R-R interval (SDNN) and power in the high-frequency (HF) band, had the most robust relationship with inflammatory markers [17]. Moreover, HRV has recently gained popularity as an indicator for explaining ANS dysfunction in the acute COVID-19 [18] or post-COVID [19] period. Although it is not an indicator of vaccine safety, HRV may provide insight into [6,7,8,9,10,11] postvaccination ANS dysfunction. However, no systematic attempt to investigate the relationship of HRV with COVID-19 vaccination or ANS dysfunction has been undertaken.

In this systematic review, we investigated the potential impact of COVID-19 vaccination on human HRV-related parameters, particularly vaccination safety, to obtain insights for the dissemination of evidence-based support for COVID-19 vaccination.

## 2. Materials and Methods

The protocol of this review has been registered in the Open Science Framework (registration number, 9UBWN). This review has been reported in accordance with the Preferred Reporting Items for Systematic Reviews and Meta-Analyses (PRISMA) statement 2020 [20] (Appendix A).

### 2.1. Eligible Criteria

#### 2.1.1. Types of Studies

Observational studies, including prospective cohort study, cross-sectional study, case–control studies, and retrospective studies, were included as was gray literature, such as conference abstracts and dissertations. Review articles and intervention studies were excluded.

#### 2.1.2. Types of Participants

Only individuals who had received COVID-19 vaccination were enrolled, without restrictions on the type of vaccine, dose, or participant’s underlying disease, sex, age, or ethnicity.

#### 2.1.3. Exposure

COVID-19 vaccination status was the only exposure that was evaluated in this analysis.

#### 2.1.4. Types of Controls

Individuals who were unvaccinated or with a prevaccination status were evaluated as controls.

#### 2.1.5. Types of Outcome Measures

The outcomes of interest were HRV-related parameters, including SDNN and HF which showed the most consistent associations with systemic inflammation and were considered the primary outcomes [17]. Secondary outcomes included all other time-domain and frequency-domain parameters, including the SDNN index, standard deviation of the average NN interval (SDANN), the square root of the mean squared differences of successive NN intervals (RMSSD), standard deviation of differences between adjacent NN intervals (SDSD), mean total power (TP), low frequency (LF), and the LF/HF ratio [16].

### 2.2. Search Strategy

One researcher (CYK) comprehensively searched four electronic bibliographic databases: MEDLINE (via PubMed), EMBASE (via Elsevier), PsycARTICLES (via ProQuest), and Cumulative Index to Nursing and Allied Health Literature (via EBSCO). Next, a manual search on Google Scholar was conducted on 29 July 2022 to identify gray literature and potentially missing documents that had been published up to the search date. Appendix A presents the search strategy and results.

### 2.3. Study Selection

Two researchers (CYK and BL) independently screened the titles and/or abstracts of the initially searched documents to identify potentially relevant studies, which was followed by full-text reviews in a two-step selection process, and studies were selected for inclusion in the review. Disagreements during the study selection process were resolved through discussion between the two researchers, and bibliographic information was managed using EndNote 20 (Clarivate Analytics, Philadelphia, PA, USA).

### 2.4. Data Extraction

The two researchers (CYK and BL) independently extracted data on the following variables from the included studies into a data extraction form (Microsoft, Redmond, WA, USA): First author, publication type, country where the study was conducted, sample size, mean age of population, types of vaccination, assessment tool of HRV parameters, HRV parameters studied and their findings, and authors’ conclusions. This form was preliminarily tested by the two researchers using two sample articles prior to the data extraction process. Any inconsistency in the data extraction between the two researchers were resolved through mutual discussion.

### 2.5. Quality Assessment

A methodological assessment was performed by using the Quality Assessment Tool for Observational Cohort and Cross-Sectional Studies developed by the National Heart, Lung, and Blood Institute group that ascertained the characteristics of the included studies and enabled a critical evaluation of the methodological quality of observational cohort studies based on 14 criteria, such as evaluation of the risk of potential for selection bias, information bias, measurement bias, or confounding bias. The quality assessment process was conducted independently by two independent researchers (CYK and BL), and any disagreements were resolved through mutual discussion.

### 2.6. Missing Data

In cases of incomplete or missing data, the authors sent an email request to the corresponding author of the study for the missing information.

### 2.7. Data Analysis

Considering the heterogeneity of the types of vaccinations and population types, including potential underlying diseases, as well as the small number of studies included in this review, a meta-analysis was not planned in the pre-registered protocol. As with this pre-planned protocol, the units of HRV parameters that were reported and the vaccines evaluated in the included studies were heterogeneous; therefore, performing a meta-analysis was unfeasible.

Nevertheless, in the supplementary file of one of the studies that included [21], detailed raw data of HRV parameters (i.e., RMSSD) for 2 weeks around the vaccination, according to vaccine types, dose of vaccination, and age and sex of the participants were reported. However, the postvaccination change in RMSSD was graphically represented only, which precluded a comparative analysis of these data with those of the other studies. Therefore, a meta-analysis of the data from only the abovementioned study was performed to ascertain the changes in the HRV parameters from the 0th day (baseline) of vaccination to that in the following 7 days and this was stratified by the types of COVID-19 vaccines. The meta-analysis was performed using the command “metan” in STATA 13.1 (StataCorp, College Station, TX, USA). Pooled data were presented as weighted mean differences (WMD) and 95% confidence intervals (CI) using a random-effects model. Both pooled mean value and pooled SD are required to operate the command “metan”. As only the mean value and 95% CI were presented [21], the SD was estimated as described in Cochrane Handbook 5.1 [22] using the algorithm in StatTools that was provided by the Department of Obstetrics and Gynecology, Chinese University of Hong Kong [23]. *p* < 0.05 indicated statistical significance.

## 3. Results

### 3.1. Study Selection

Among the initially searched 376 documents, 116 duplicate publications were removed; the title and abstract of the remaining 260 documents were screened initially, and 249 irrelevant documents were excluded. The full-texts of the remaining 11 documents were reviewed; then, two opinion articles [24,25], three on the impact of SARS-CoV-2 infection [26,27,28], and one on human rhinovirus [29] were excluded, and the remaining five observational studies [21,30,31,32,33] were included in this review (Figure 1).

### 3.2. Characteristics of the Included Studies

The five included studies were all journal articles; four [21,30,31,32] were conducted in the United States, and the remaining study [33] was conducted in Israel. Four studies [30,31,32,33] were prospective cohort studies, and one [21] was a retrospective cohort study. Three of the included studies [21,32,33], had more than 10,000 participants. Three studies [30,31,33] investigated the impact of Pfizer-BioNTech COVID-19 vaccination on HRV parameters. Mason et al. [32] investigated the effects of Pfizer-BioNTech, Moderna, and Johnson & Johnson-Janssen (J/J&J) COVID-19 vaccinations; Presby et al. [21] investigated the impacts of AstraZeneca, J/J&J, Moderna, or Pfizer/BioNTech COVID-19 vaccinations. To assess the HRV parameters of participants, three studies [21,30,33] used a commercial wrist-worn sensor, one [32] used a commercial finger sensor, and the remaining study [31] used an FDA-approved chest-patch sensor (Table 1).

### 3.3. Methodological Quality Assessment

The research questions (Q1) and populations (Q2) of the five included studies [21,30,31,32,33] were clearly described. The participation rate of eligible persons (Q3) in a study [21] was not described clearly. In the study of Mason et al. [32], the number of eligible persons was 2401, although the number of participants was 1179 (49.1%). The participation rate of other studies [30,31,33] were more than 50%. In the five studies [21,30,31,32,33], the participants were recruited from similar populations (Q4). Except for the study by Gepner et al. [31], the studies [21,30,32,33] did not justify their sample size or describe their sample-size calculation method (Q5). In the five studies [21,30,31,32,33], the exposures of interest (i.e., vaccinations) were measured prior to the measurement (Q6) of outcomes (i.e., HRV parameters), examined different levels of the exposures (i.e., dose of vaccinations) (Q8), and clearly described the exposure measures (Q9) and the outcome measures (Q11). Moreover, as the parameters were known to be affected even immediately after the vaccination, all the studies [21,30,31,32,33] had a sufficient timeframe for determining an association between vaccination and HRV parameters (Q7). As the exposures do not need to be assessed more than once over time, Q10 for the five studies [21,30,31,32,33] was assessed as inapplicable. In the study of Hajduczok et al. [30], the assessor was blinded to the participants’ exposure (Q12); the other studies [21,31,32,33] did not report assessor-blinded procedures. As two studies did not perform [21,33] longitudinal assessments, Q13 was inapplicable for these studies; the remaining three studies [30,31,32] had 20% or less loss to follow-up. In all five studies [21,30,31,32,33], key potential confounding variables were not measured or adjusted statistically in their analysis (Q14). Although Presby et al. [21] conducted analysis according to some confounders, including age and sex, none of the studies measured and/or adjusted for major cardiovascular diseases, such as heart failure (Table 2).

### 3.4. Impact of COVID-19 Vaccinations on HRV Parameters

#### 3.4.1. Qualitative Analysis

Except for one study that provided a proxy indicator of HRV (i.e., Garmin’s stress level [33]), all other studies used RMSSD as the main HRV parameter [21,30,31,32]. Presby et al. [21] reported the RMSSD, whereas the remaining studies provided only the change in RMSSD [30,31] or the Spearman’s rank order correlation coefficient [32] of RMSSD. Baseline data differed among the included studies. In three studies [30,32,33], the criteria were recorded for a period, whereas the remaining two studies [21,31] recorded the criteria at a certain timepoint. The criteria in Hajduczok et al. [30], Mason et al. [32], and Mofaz et al. [33] were recorded at least 24 of 45 days (Day-45 to Day-1), 14 to 4 days (Day-14 to Day-4), and 7 to 1 day (Day-7 to Day-1) prior to vaccination, respectively. In the studies of Gepner et al. [31] and Presby et al., the criteria were recorded the day prior to the dose (Day-1) and 1 week before the dose (Day-7), respectively [21].

Hajduczok et al. [30] reported an RMSSD decrease in their participants on the day after the first dose of Pfizer-BioNTech COVID-19 vaccination (mean percent change, −13.44%; SD, 13.62%) that normalized on Day 3 and remained stable. With the second dose, RMSSD decreased on the day after the vaccination (mean percent change, −9.25%; SD, 22.69%), but quickly normalized on Day 2 and remained stable. Gepner et al. [31] found that the RMSSD (per ten beats, %) decreased on the day after the second dose of the Pfizer-BioNTech COVID-19 vaccination (during the day: from 6.89 ± 0.26% to 0.01 ± 4.07%; at night: from 4.78 ± 0.23% to 0.13 ± 4.00%), but these changes faded from Day 2 and the RMSSD returned to the baseline level (during the day: 7.71 ± 5.41%; at night: 13 ± 10%).

Using Spearman rank order correlations between the SARS-CoV-2 receptor-binding domain (RBD) antibody responses and the relevant metrics, Mason et al. [32] investigated the clinical implications of changes in some physiological metrics, including sleep status, heart rate, respiratory rate, temperature deviation, and HRV (i.e., RMSSD) after vaccinations with the Pfizer-BioNTech, Moderna, and J/J&J COVID-19 vaccines. Overall, significant relationships in these metrics (i.e., decreased RMSSD and higher RBD antibody response) were observed for the second dose of the two mRNA vaccines (Moderna and Pfizer-BioNTech’s COVID-19 vaccines). In the adjusted multivariate models, RMSSD decreased in the night immediately after the second dose of the two mRNA vaccines, but did not independently predict a higher RBD antibody response (*p* = 0.191) and only increased the temperature deviation (*p* < 0.001). In the studies of Mason et al. [32] and Presby et al. [21], differences according to the type of vaccine were detected wherein the first dose of the AstraZeneca vaccine and the second doses of Moderna and Pfizer vaccines induced greater changes in physiological parameters, including HRV (all, *p* < 0.001), compared to their respective first or second dose. Interestingly, the postvaccination changes in RMSSD were more pronounced in women than in men (*p* < 0.001) and in younger groups (i.e., 18- to 29-year-olds) than in the older group (i.e., ≥55 years) (*p* = 0.02 to <0.001). These changes disappeared and the values returned to baseline levels within 4 nights after the vaccination.

Mofaz et al. [33] observed changes in the HRV-based stress indicator (i.e., Garmin’s stress level) after the first, second, and booster doses of the Pfizer-BioNTech COVID-19 vaccine. Especially, considerable increases in heart rate and the stress indicator were detected during the first 48 h after the booster administration (*p*-value not reported). Compared to the changes in the heart rate (*p* = 0.004) and Garmin’s stress level (*p* < 0.001) after the first dose, the changes after the third dose were significantly greater.

#### 3.4.2. Quantitative Analysis

A meta-analysis was performed by reconstructing the raw data presented in the study of Presby et al. [21], the postvaccination value of RMSSD (ms) was compared with the baseline value (Day 0), and this showed that RMSSD decreased significantly on Day 1 after the first dose of the AstraZeneca vaccine (WMD, −13.37; 95% CI, −14.86 to −11.88). Thereafter, no significant difference with baseline values was observed for the remainder of the observation period, except that the RMSSD value was slightly higher than the baseline on Day 3 (WMD, 2.04; 95% CI, 0.28–3.80; Figure 2a). A similar trend of non-significant difference was observed after the second dose of the AstraZeneca vaccine. Thus, the RMSSD value decreased, compared to the baseline value, on Day 1 of the vaccination (WMD, −5.56; 95% CI, −11.25 to 0.3), although a statistically significant difference was not achieved (Figure 2b). The RMSSD value significantly decreased on Day 1 after the first dose of Moderna vaccination on Day 1 (WMD, −3.95; 95% CI, −4.54 to −3.36) and Day 2 (WMD, −4.10; 95% CI, −4.81 to −3.40), compared to that at the baseline. No significant difference from baseline was observed on other days until the 7th day after the vaccination (Figure 2c). A similar trend was observed at the second dose of Moderna vaccine: RMSSD values on days 1 (WMD, −13.84; 95% CI, −14.53 to −13.14) and 2 (WMD, −3.16; 95% CI, −3.81 to −2.51) significantly decreased compared to those at the baseline. Thereafter, although the RMSSD value on Day 3 (WMD, 1.22; 95% CI, 0.61–1.83) was significantly higher than that at baseline, the values on days 5 (WMD, −0.74; 95% CI, −1.43 to −0.06), 6 (WMD, −0.93; 95% CI, −0.54 to −0.32), and 7 (WMD, −1.24; 95% CI, −1.96 to −0.52) significantly decreased compared to that at the baseline (Figure 2d). In case of the first dose of Pfizer/BioNTech vaccination, RMSSD values on days 1 (WMD, −2.28; 95% CI, −2.79 to −1.76), 2 (WMD, −1.63; 95% CI, −2.19 to −1.07), 3 (WMD, −0.62; 95% CI, −1.20 to −0.04), 4 (WMD, −0.54; 95% CI, −1.07 to −0.00), 5 (WMD, −0.78; 95% CI, −1.28 to −0.29), and 7 (WMD, −0.56; 95% CI, −1.10 to −0.01) were significantly decreased as compared to those at the baseline (Figure 2e). A similar trend was observed at the second dose of Pfizer/BioNTech vaccination. That is, compared to those of baseline, RMSSD values on days 1 (WMD, −7.54; 95% CI, −8.10 to −6.99), 2 (WMD, −0.98; 95% CI, −1.52 to −0.45), 5 (WMD, −1.17; 95% CI, −1.71 to −0.62), 6 (WMD, −0.80; 95% CI, −1.29 to −0.31), and 7 (WMD, −1.15; 95% CI, −1.65 to −0.65) were significantly decreased (Figure 2f). In case of the single dose of J/J&J vaccination, compared to baseline, RMSSD value significantly decreased on day 1 (WMD, −16.93; 95% CI, −18.42 to −15.45). Thereafter, the RMSSD value was significantly higher than the baseline on days 2 (WMD, 2.26; 95% CI, 0.86–3.66), 3 (WMD, 3.34; 95% CI, 1.97–4.70), and 4 (WMD, 2.01; 95% CI, 0.47–3.55; Figure 2g). Regardless of vaccine type and dose, a meta-analysis of differences of RMSSD value from baseline according to days after vaccination showed high heterogeneity 1–3 days after vaccination (*I^2^* = 85.2–99.4%), which decreased 4 days after vaccination (*I^2^* = 44.1–66.6%) (Appendix A). Besides the meta-analysis, the rate of change of RMSSD from baseline (∆RMSSD (%)) in Presby et al. [21] was calculated and visualized together with the data of Hajduczok et al. [30] on ∆RMSSD (%). Analysis of ∆RMSSD (%) up to the 6th [30] or 7th day [21] after the first and second doses of Pfizer/BioNTech vaccination showed a rapid change within the first 3 days of vaccination in both cases, and a return to the baseline thereafter (Figure 3).

### 3.5. Relationship between HRV Parameters and Adverse Events after COVID-19 Vaccinations

Gepner et al. [31] found a short-term decrease of RMSSD value after COVID-19 vaccinations, even in presumably asymptomatic participants. However, changes of the value in symptomatic participants were significantly higher than those in asymptomatic participants during the first night post vaccination (*p* < 0.05). Furthermore, Mofaz et al. [33] reported similar changes of the HRV-based stress indicator (i.e., Garmin’s stress level; *p* = 0.54) for the trends of postvaccination physical symptoms of the participants. The self-reported reactions after the third dose were significantly greater than that after the first dose (*p* < 0.001). Lastly, Presby et al. [21] found that fever or chills reported by the participants were associated with the deviations of RMSSD value (β = −7.3 ± 1.0). The authors found significant postvaccination changes in some physiological parameters of asymptomatic participants, including changes in the resting heart rate and percent sleep derived from light sleep (both, *p* < 0.001), but not in the RMSSD (*p* > 0.05).

## 4. Discussion

### 4.1. Findings of This Review

This systematic review was performed to investigate the impact of COVID-19 vaccinations on human HRV parameters. Through comprehensive searches, five observational studies [21,30,31,32,33] were identified and included in this review. According to the findings of this review, COVID-19 vaccinations may be related to a short-term decrease of the RMSSD value. Some studies suggested that the decrease was related to self-reported reactions after the vaccinations [21,31,33]. However, the results of HRV changes after the vaccinations were mixed in asymptomatic participants [21,31]. Moreover, some studies indicated different impacts on the HRV parameter according to the types of vaccines as well as the doses. Mason et al. [32] found that only the second dose of the Moderna and Pfizer-BioNTech vaccinations were significantly related to the SARS-CoV-2 RBD antibody responses, whereas the J/J&J vaccination was not. Moreover, Presby et al. [21] found that the first dose of the AstraZeneca vaccine, compared to the second dose, and the second doses of Moderna and Pfizer vaccines compared to the first doses, showed greater changes in HRV. Finally, Mofaz et al. [33] found that the third (booster) dose of Pfizer-BioNTech had a bigger impact on the HRV-based stress indicator, as compared to the first dose. Moreover, Presby et al. [21] reported that the impact of COVID-19 vaccinations on the RMSSD was greater in women than in men, and in the younger group than in the older group. The methodological quality of the included studies was not optimal. Especially, as none of the studies statistically measured and adjusted the key potential confounding variables.

### 4.2. Clinical Interpretation

The results of this review confirm that changes in HRV after COVID-19 vaccine show significant short-term changes, for up to 3 days, and rapidly return to baseline levels. However, the findings may be contradictory to the results of some case reports of persistent adverse reactions following COVID-19 vaccination [6,7,8,9,10], including postural orthostatic tachycardia syndrome (POTS) after COVID-19 vaccinations [6,7,8,11,34]. Significant changes in HRV parameters, including RMSSD, HF, and LF/HF ratio, have been documented in patients with POTS [35]; however, this review supports the overall safety of COVID-19 vaccination in terms of HRV parameters. Therefore, the findings of this review support the assumption that POTS following COVID-19 vaccination is not a common phenomenon and may be an individualized specific response.

As the studies included in this review investigated the population-level changes in HRV parameters, particularly RMSSD, following COVID-19 vaccination a, individual-specific responses may be overshadowed. Although the causal mechanism is unclear, Eldokla et al. [34] reported that serum markers of possible autoimmunity were found in 4 out of 5 cases with POTS following mRNA COVID-19 vaccination in their case series, indicating that a vaccine-induced autoimmune response was a potential mechanism of POTS after the vaccinations; however, the authors indicated that these patients did not require immunotherapy or intravenous immunoglobulin [34]. Some cases of POTS have been reported in COVID-19 patients and Blitshteyn et al. [11] reported that, of their 20 POTS cases following SARS-CoV-2 infection, 20% had elevated autoimmune or inflammatory markers. Autoimmunity and the presence of autoantibodies are mechanisms underlying autonomic dysfunction following HPV vaccination [12]. Therefore, the risk of POTS should be reviewed from the perspective of individual vulnerability, such as autoimmunity, and concerns about postvaccination POTS should not hinder COVID-19 vaccination in the general public.

Rarely, cases of MIS after the COVID-19 AstraZeneca vaccination have been reported [15]. Park et al. found a decrease in natural killer cell activity in the patient with MIS after COVID-19 vaccination, and hypothesized that the pathophysiology of MIS was related to abnormal activation of innate immunity [15]. Given the known association between inflammation and HRV parameters [17], the findings of this review support the theory that MIS after COVID-19 vaccination may not be a generalized concomitant risk. However, according to the results of a literature review of MIS after COVID-19 vaccination, the onset of symptoms in some cases occurred within a few hours, whereas in some other cases, these effects occurred 2 to 7 weeks after the vaccinations [17]. Therefore, the findings of this review, which failed to analyze the long-term stability of HRV parameters, do not interpret adverse events that potentially occur at a delayed timepoint following COVID-19 vaccination. As HRV parameters do not fully represent ANS and can be influenced by other factors, including respiration [16], the limitation that the stability of HRV parameters after the vaccinations cannot be interpreted as indicating the stability of ANS activity.

### 4.3. Limitations

This systematic review is the first comprehensive investigation of the association between COVID-19 vaccination and HRV parameters. Research on the adverse reactions of COVID-19 vaccination are scarce, and in a situation where COVID-19 vaccination hesitancy is an issue [2], it is meaningful from the public health perspective to investigate HRV: an objective parameter of the post-COVID-19 vaccination clinical status. However, this review has the following limitations. First, the included studies reported mainly changes in RMSSD among HRV parameters after COVID-19 vaccination. RMSSD is a time-domain parameter that is frequently used to quantify HRV and reflects the integrity of vagus nerve-mediated autonomic control of the heart, and its association with sudden unexplained death, cardiovascular diseases, and all-cause mortality has been reported [36,37]. However, there was a limitation in linking the findings to clinical interpretation because of a lack of reports of some parameters that are important for the clinical interpretation of HRV, such as SDNN and HF, which are the primary outcomes of this review. Second, the included studies reported HRV responses at the group level, rather than individual-specific responses; therefore, individual specificities, including autoimmune response and allergic reactions [38], may have been masked. Although anaphylactic reactions from SARS-CoV-2 vaccine are known to be rare [39], individual-specific responses to COVID-19 vaccination and vulnerable populations should be further elucidated. Third, as the number of studies included in this study was small and heterogeneity in the unit of reported parameters exists, a meta-analysis to synthesize the data of the studies was not possible. However, a meta-analysis was feasible using data in a study by Presby et al. [21], which reported data from more than 50,000 participants. Fourth, limitations in the reliability of measurements of the HRV parameters reported in the included studies should be acknowledged. Among the included studies, only one study [31] used an FDA-approved highly sensitive chest-patch sensor, whereas the remaining studies [21,30,32,33] used commercially available wearable sensors. Future studies will need to increase the reliability of HRV measurements to better demonstrate the association between HRV and COVID-19 vaccination. Fifth, the included studies focused on the stability of short-term HRV parameters after COVID-19 vaccination; none of the studies measured long-term changes. Thus, the results of these studies show that changes in RMSSD are normalized within 3 days after COVID-19 vaccination; however, the findings do not eliminate the possibility of delayed adverse events. For example, given that adverse events such as MIS have been reported 2 to 7 weeks after that vaccination [17], the stability of long-term HRV parameters after COVID-19 vaccination may be an attractive research topic for future studies. Sixth, the heterogeneity of the baseline data of the included studies is a major limitation of this review. Although the included studies considered data within 1.5 months of vaccination as the baseline data, the heterogeneity could not be ignored. The fact that baseline data were obtained within 1 week before vaccination in the two studies [21,33] that included the largest sample among the included studies can be considered as an important implication for future research on this topic. Seventh, a major limitation of the included studies is that the studies did not measure and/or adjust for some important confounders such as cardiovascular diseases and the impact of the COVID-19 lockdowns. Given that HRV is an important indicator of cardiac health [40], and that individuals experiencing lockdown in the context of COVID-19 [41] or with cardiopathy [42] show significantly different outcomes on the HRV index compared to controls, future research could consider experiencing lockdown and history of cardiovascular disease as their covariate. Finally, since this review focused on changes of HRV after COVID-19 vaccination, the results do not guarantee the safety of the COVID-19 vaccine, but only suggest some safety aspects related to COVID-19 vaccination. Nevertheless, the findings of the association between COVID-19 vaccination and HRV provide an insight into the context of ANS dysfunction after COVID-19 vaccination [6,7,8,11,34].

## 5. Conclusions

The findings of this review reveal short-term changes in HRV parameters, particularly RMSSD, following COVID-19 vaccination. In general, this short-term change normalized within 3 days; there were interparticipant differences, including the vaccine type and dose, and the participant’s sex and age. However, in the included studies, other important parameters, besides RMSSD, were not reported, and the limitation remains that the long-term HRV stability after the vaccinations was not reported. The results of this review contribute to the understanding of COVID-19 vaccine safety from the perspective of evidence-based medicine, and have public health implications for alleviating COVID-19 vaccine hesitancy that impedes vaccine uptake by the general population.

## Figures and Tables

**Figure 1 vaccines-10-02095-f001:**
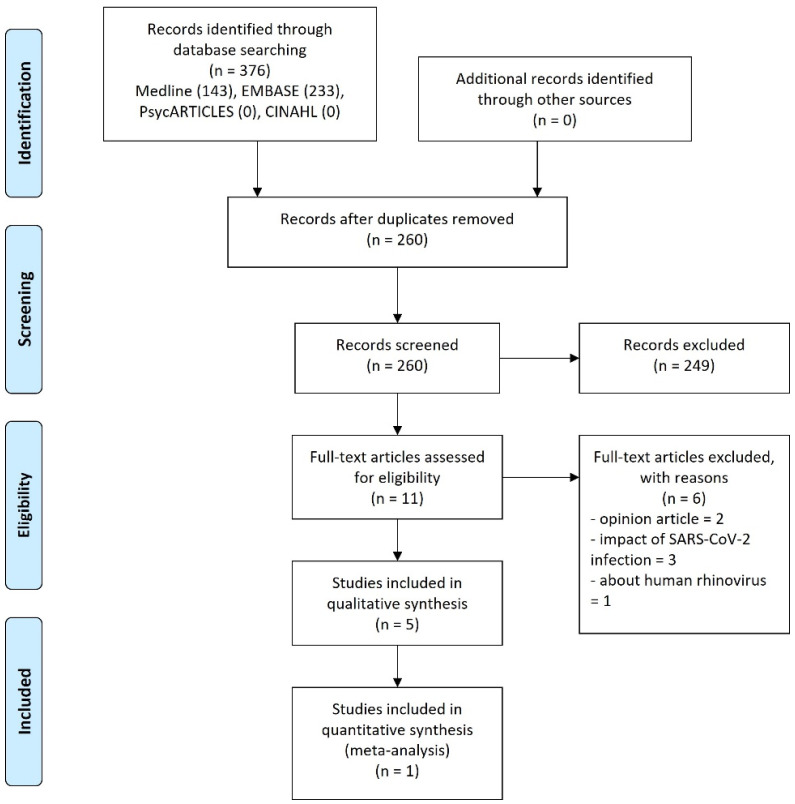
PRISMA flow diagram of this review. Abbreviations. CINAHL, Cumulative Index to Nursing and Allied Health Literature; SARS-CoV-2, severe acute respiratory syndrome coronavirus 2.

**Figure 2 vaccines-10-02095-f002:**
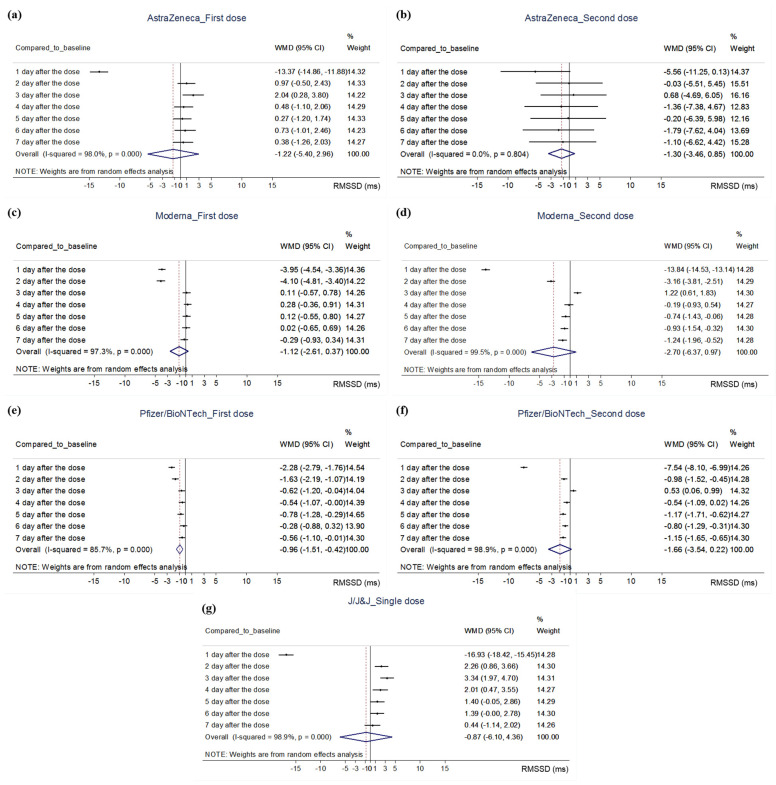
Meta-analysis on the data from Presby et al. (2022). Abbreviations. CI, confidence interval; J/J&J, Johnson & Johnson’s Janssen; RMSSD, the square root of the mean squared differences of successive NN intervals; WMD, weighted mean difference. Note. (**a**) the impact of the first dose of the AstraZeneca vaccine on RMSSD, (**b**) the impact of second dose of the AstraZeneca vaccine on RMSSD, (**c**) the impact of first dose of the Moderna vaccine on RMSSD, (**d**) the impact of the second dose of the Moderna vaccine on RMSSD, (**e**) the impact of the first dose of the Pfizer/BioNTech vaccine on RMSSD, (**f**) the impact of the second dose of the Pfizer/BioNTech vaccine on RMSSD, (**g**) the impact of single dose of J/J&J vaccine on RMSSD.

**Figure 3 vaccines-10-02095-f003:**
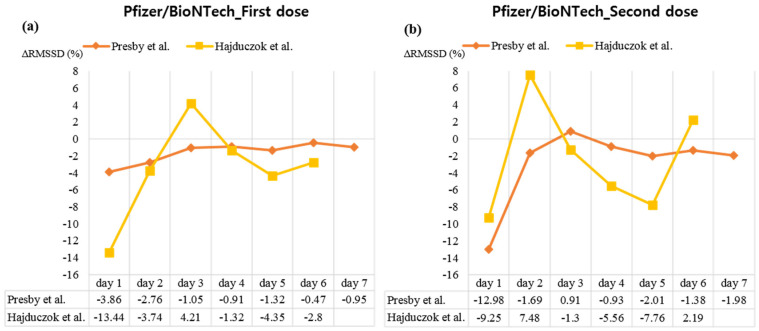
Changes of heart rate variability from baseline after Pfizer/BioNTech vaccinations. Abbreviations. RMSSD, the square root of the mean squared differences of successive NN intervals. Note. (**a**) the impact of the first dose of the Pfizer/BioNTech vaccine on RMSSD, (**b**) the impact of second dose of the Pfizer/BioNTech vaccine on RMSSD. The closer the value is to 0 on the Y-axis, the closer it is to the baseline value.

**Table 1 vaccines-10-02095-t001:** Characteristics of included studies.

Author (Country)	Study Type	Sample Size (M:F)	Population (Mean Age, Yr)	Type of Vaccinations (Dose)	Assessment Tool for HRV	HRV Parameters (Unit)
Hajduczok 2021 (U.S.)	prospective cohort study	19 (9:10)	adult (28.8 ± 2.2)	Pfizer-BioNTech COVID-19 vaccine (first and second doses)	commercial wearable sensor on wrist (WHOOP Strap)	1. RMSSD (change, %)
Gepner 2022 (U.S.)	prospective cohort study	160 (70:90)	adult (median: 40, range: 21–78)	Pfizer-BioNTech COVID-19 vaccine (second dose)	FDA-approved chest-patch sensor	1. RMSSD (per ten beats, %)
Mason 2022 (U.S.)	prospective cohort study	1. Pfizer-BioNTech: 706 (334:371, others: a participant classified as intersex) 2. Moderna: 366 (166:200) 3. J/J&J: 107 (53:54)	1. 50.4 ± 11.4 2. 52.8 ± 12.0 3. 49.5 ± 9.9	Pfizer-BioNTech, Moderna, J/J&J (first and second doses, except for J/J&J)	commercial wearable sensor on finger (Oura Ring)	1. RMSSD (Spearman’s rank order correlation coefficient)
Mofaz 2022 (Israel)	prospective cohort study	1609 (755:854)	adult (median: 52, range: 18–88)	Pfizer-BioNTech COVID-19 vaccine (first, second, and booster doses)	commercial wearable sensor on wrist (Garmin Vivo-smart 4 smartwatches)	1. Garmin’s stress level (HRV-based stress indicator)
Presby 2022 (U.S.)	retrospective analysis	first dose/second dose 1. AstraZeneca: 3457 (870:2587) / 325 (86:239) 2. J/J&J: 4584 (1416:3168) 3. Moderna: 17,632 (6063:11,569)/16,987 (6025:10,962) 4. Pfizer/BioNTech: 29,366 (10,027:19,339)/27,084 (9614:17,470)	adult (18 or older)	AstraZeneca, J/J&J, Moderna, or Pfizer/BioNTech vaccine (not limited)	commercial wearable sensor on wrist (WHOOP Strap)	1. RMSSD (ms)

Abbreviations. COVID-19, Coronavirus disease of 2019; FDA, U.S. Food and Drug Administration; HRV, heart rate variability; J/J&J, Johnson & Johnson’s Janssen; RMSSD, the square root of the mean squared differences of successive NN intervals.

**Table 2 vaccines-10-02095-t002:** Methodological quality of included studies.

Author	Q1	Q2	Q3	Q4	Q5	Q6	Q7	Q8	Q9	Q10	Q11	Q12	Q13	Q14
Hajduczok 2021	Y	Y	Y	Y	N	Y	Y	Y	Y	NA	Y	Y	N	N
Gepner 2022	Y	Y	Y	Y	Y	Y	Y	Y	Y	NA	Y	NR	N	N
Mason 2022	Y	Y	N	Y	N	Y	Y	Y	Y	NA	Y	NR	N	N
Mofaz 2022	Y	Y	Y	Y	N	Y	Y	Y	Y	NA	Y	NR	NA	N
Presby 2022	Y	Y	NR	Y	N	Y	Y	Y	Y	NA	Y	NR	NR	N

Abbreviation. N, no; NA, not applicable; NR, not reported; Y, yes. Note. Q1. Was the research question or objective in this paper clearly stated?; Q2. Was the study population clearly specified and defined?; Q3. Was the participation rate of eligible persons at least 50%?; Q4. Were all the subjects selected or recruited from the same or similar populations (including the same time period)? Were inclusion and exclusion criteria for being in the study prespecified and applied uniformly to all participants?; Q5. Was a sample size justification, power description, or variance and effect estimates provided?; Q6. For the analyses in this paper, were the exposure(s) of interest measured prior to the outcome(s) being measured?; Q7. Was the timeframe sufficient so that one could reasonably expect to see an association between exposure and outcome if it existed?; Q8. For exposures that can vary in amount or level, did the study examine different levels of the exposure as related to the outcome (e.g., categories of exposure, or exposure measured as continuous variable)?; Q9. Were the exposure measures (independent variables) clearly defined, valid, reliable, and implemented consistently across all study participants?; Q10. Was the exposure(s) assessed more than once over time?; Q11. Were the outcome measures (dependent variables) clearly defined, valid, reliable, and implemented consistently across all study participants?; Q12. Were the outcome assessors blinded to the exposure status of participants?; Q13. Was loss to follow-up after baseline 20% or less?; Q14. Were key potential confounding variables measured and adjusted statistically for their impact on the relationship between exposure(s) and outcome(s)?

## Data Availability

This data used to support the findings of this study are included within the article.

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
