# Peer review of "Impact of COVID-19 Vaccination on Heart Rate Variability: A Systematic Review"

_vaccines, 2022, doi:10.3390/vaccines10122095_

Round 1

Reviewer 1 Report

This is an interesting study reviewing 5 studies for HRV of vaccination. In the comparison, it is not clear whether the baseline of HRV changes for different studies was similar. For example, even though both studies report the percent change in RMSSD, if one takes the 20 days before vaccination as the baseline whereas another study only takes 3 days before vaccination as the baseline, then the changes could be somewhat less reliable in the latter case. Perhaps such differences in protocol also need to be considered in the comparisons.

Line 237: For question 14: Were key potential confounding variables measured and adjusted statistically for their impact on the relationship between exposure(s) and outcome(s)?    All the studies were rated "No", perhaps the author should list some of the confounding variables (perceived to be important) to guide future studies in the discussion section. Possible confounding factors could be age, gender etc but the study of Presby did include analyses based on age groups, so perhaps there are additional confounding factors that the authors have in mind, and they could explain further about these other factors.

Figure 2 could be most easily understood if the data is also represented by a table, showing the types of vaccines on the Y axis, and the first or second dose on the X axis, or vice versa. In any case, the figure is not very clear, it would be better to show the types of vaccines on the figure itself so that one can read the results quickly.

Author Response

  • Response to Comments from Reviewer 1

Overall comment:

This is an interesting study reviewing 5 studies for HRV of vaccination.

Response:

Thank you for your careful review and insightful comments that have significantly enhanced our manuscript.

Comment 1:

In the comparison, it is not clear whether the baseline of HRV changes for different studies was similar. For example, even though both studies report the percent change in RMSSD, if one takes the 20 days before vaccination as the baseline whereas another study only takes 3 days before vaccination as the baseline, then the changes could be somewhat less reliable in the latter case. Perhaps such differences in protocol also need to be considered in the comparisons.

Response 1:           

Thank you for the comments. We completely agree that the differences in the criteria applied to the baseline data may be an important factor in interpreting the results of the included studies. Therefore, we have revised the manuscript to further describe the criteria applied to the baseline data from the included studies; moreover, the heterogeneity of the included studies has been described as one of the major limitations of this review.

“Baseline data differed among the included studies. In three studies [30,32,33], the criteria were recorded for a period, whereas the remaining two studies [21,31] recorded the criteria at a certain timepoint. The criteria in Hajduczok et al. [30], Mason et al. [32], and Mofaz et al. [33] were recorded at least 24 of 45 days (Day −45 to Day −1), 14 to 4 days (Day −14 to Day −4), and 7 to 1 day (Day −7 to Day −1) prior to vaccination, respectively. In the studies of Gepner et al. [31] and Presby et al., the criteria were recorded the day prior to the dose (Day −1) and 1 week before the dose (Day −7), respectively [21].”

(Please refer red words in page 8)

Sixth, the heterogeneity of the baseline data of the included studies is a major limitation of this review. Although the included studies considered data within 1.5 months of vaccination as the baseline data, the heterogeneity could not be ignored. The fact that baseline data were obtained within 1 week before vaccination in the two studies [21,33] that included the largest sample among the included studies can be considered as an important implication for future research on this topic.”

(Please refer red words in page 13)

Comment 2:

Line 237: For question 14: Were key potential confounding variables measured and adjusted statistically for their impact on the relationship between exposure(s) and outcome(s)?    All the studies were rated "No", perhaps the author should list some of the confounding variables (perceived to be important) to guide future studies in the discussion section. Possible confounding factors could be age, gender etc but the study of Presby did include analyses based on age groups, so perhaps there are additional confounding factors that the authors have in mind, and they could explain further about these other factors.

Response 2:           

Thank you for the comments. In the revised manuscript, we have added descriptions of potential confounding variables, such as lockdown in the context of COVID-19 and major cardiovascular disease such as heart failure, that could have had a significant impact on the HRV outcomes. We concur with your insight that this limitation may constitute important implications for future research in this field.

“In all five studies [21,30-33], key potential confounding variables were not measured or adjusted statistically in their analysis (Q14). Although Presby et al. [21] conducted analysis according to some confounders, including age and sex, none of the studies measured and/or adjusted for major cardiovascular diseases, such as heart failure (Table 2).”

(Please refer red words in page 7)

Seventh, a major limitation of the included studies is that the studies did not measure and/or adjust for some important confounders such as cardiovascular diseases and the impact of the COVID-19 lockdowns. Given that HRV is an important indicator of cardiac health [40], and that individuals experiencing lockdown in the context of COVID-19 [41] or with cardiopathy [42] show significantly different outcomes on the HRV index compared to controls, future research could consider experiencing lockdown and history of cardiovascular disease as their covariate.”

(Please refer red words in page 13)

Comment 3:

Figure 2 could be most easily understood if the data is also represented by a table, showing the types of vaccines on the Y axis, and the first or second dose on the X axis, or vice versa. In any case, the figure is not very clear, it would be better to show the types of vaccines on the figure itself so that one can read the results quickly.

Response 3:           

Thank you for the comments. We have addressed the issue indicated by this comment by improving the legibility and comprehensibility of Figure 2 through the following major changes.

1) The types of vaccine and dose are specified in the title of each figure part from (a) to (g).

2) The font size of the text has been increased.

3) The measurement unit has been added for the values indicated on the x-axis in the figure.

4) We have rearranged figure parts (a) to (g) to ensure that the data for the same vaccine type are placed in the same column.

We have not added the table, which was suggested by the reviewer, to avoid redundancy by restating the same data that has been reported in the figures. However, we will add the table if you consider this revised figure difficult to understand. Therefore, we welcome the reviewer's comments.

Reviewer 2 Report

1. At first, I have to bring up the writing problem. The English is OK and there aren’t many mistakes in grammar, but the writing skills are very poor. For example, “which started at the end of 2019” of line 29 is redundant, as well as “thus, the development of COVID-19 vaccines is ongoing” in line 35 and “with regard to COVID-19 vaccinations” in line 39. I will stop here, but I strongly suggest the authors put more efforts in the writing.

2. In section 2.4 line 127, the phrase “pilot-tested Excel file” can be confusing, please explain.

3. There is a typo in Table 1, row 4 column 3. Total size 706 not equal 334M + 371F.

And column 4 of the last row (age range of the work of Presby) is incomplete or incorrect.

4. The text size of Figure 2 should be optimized, for it is very difficult to see clearly right now.

5. The quantitative analysis based on the data from Presby’s work (section 3.4.2) makes a great portion of this review. However, the same analysis had been already done by that group. There is a great resemblance between Figure 2 of this review and Figure 2C of the Presby paper. More and deeper data digging works are needed.

6. (This question need not to be answered)

The purpose of this paper is to “have public health implications in alleviating the hesitancy associated with COVID-19 vaccine uptake by the general population”. Generally speaking, the most concerned topics in this field are allergic reactions and side-effects, as well as the long period safety issue. When focusing on the side-effects, HRV plays a much unimportant role compared with other symptoms, so a lot of related research did not measure it at all, that is why there is so little research data. And for the long period safety, this paper did not talk about it at all. So I can not say the purpose of this paper is well achieved. Still, combining HRV with COVID-19 vaccine safety issues is a very interesting point, the author just need to put more effort in it (narrow down and focus on the main topic, digging the data in a deeper level or another dimension, rephrase some section such as section 2 and 4 to reduce the length of this paper).

Author Response

  • Response to Comments from Reviewer 2

Comment 1:

  1. At first, I have to bring up the writing problem. The English is OK and there aren’t many mistakes in grammar, but the writing skills are very poor. For example, “which started at the end of 2019” of line 29 is redundant, as well as “thus, the development of COVID-19 vaccines is ongoing” in line 35 and “with regard to COVID-19 vaccinations” in line 39. I will stop here, but I strongly suggest the authors put more efforts in the writing.

Response 1:

Thank you for the comments. First, we would like to apologize for any inconvenience that was caused to the reviewer while reviewing this manuscript. To address the issues indicated in your feedback, we have ensured that the revised manuscript has been thoroughly reviewed by a professional English editing company as well as by the authors.

Changed expressions are marked with red words throughout the manuscript. A certificate of the academic editing manuscript is attached.

In addition, the ungrammatical expression of the title of this manuscript was also pointed out during the editing process, so it was changed to ‘Impact of COVID-19 vaccination on heart rate variability: a systematic review’, from ‘Do COVID-19 vaccinations impact on human heart rate variability? a systematic review’.

Comment 2:

  1. In section 2.4 line 127, the phrase “pilot-tested Excel file” can be confusing, please explain.

Response 2:

Thank you for the comments. The potentially confusing term has been modified as follows.

“The two researchers (CYK and BL) independently extracted data on the following variables from the included studies into a data extraction form (Microsoft, Redmond, WA, USA): First author, publication type, country where the study was conducted, sample size, mean age of population, types of vaccination, assessment tool of HRV parameters, HRV parameters studied and their findings, and authors’ conclusions. This form was preliminarily tested by the two researchers using two sample articles prior to the data extraction process. Any inconsistency in the data extraction between the two researchers were resolved through mutual discussion.”

(Please refer red words in page 3)

Comment 3:

  1. There is a typo in Table 1, row 4 column 3. Total size 706 not equal 334M + 371F.

And column 4 of the last row (age range of the work of Presby) is incomplete or incorrect.

Response 3:

Thank you for the comments. The lack of clarity is because one of the participants was classified as intersex, as neither male nor female. We have added this information in the table. In this table, “18-” refers to age ≥18 years, and information on the maximum age of the participants was not reported. Therefore, we have included this information in the revised manuscript as follows.

1) Pfizer-BioNTech: 706 (334:371, others: a participant classified as intersex)

2) adult (18 or older)

(Please refer red words in page 6)

Comment 4:

  1. The text size of Figure 2 should be optimized, for it is very difficult to see clearly right now.

Response 4:

Thank you for the comments. We have addressed the issue indicated by this comment by improving the legibility and comprehensibility of Figure 2 through the following major changes.

1) The types of vaccine and dose are specified in the title of each figure part from (a) to (g).

2) The font size of the text has been increased.

3) The measurement unit has been added for the values indicated on the x-axis in the figure.

4) We have rearranged figure parts (a) to (g) to ensure that the data for the same vaccine type are placed in the same column.

Comment 5:

  1. The quantitative analysis based on the data from Presby’s work (section 3.4.2) makes a great portion of this review. However, the same analysis had been already done by that group. There is a great resemblance between Figure 2 of this review and Figure 2C of the Presby paper. More and deeper data digging works are needed.

Response 5:

Thank you for the comments. Instead of merely repeating the existing work of Presby et al., we have made the following efforts to derive and interpret the data to present the outcomes of a deeper analysis in the revised manuscript.

1) By calculating the values that were not presented in the paper by Presby et al. (presented as an image only), it was possible to compare the results with those of another included study (i.e., Hajduczok et al.).

“Nevertheless, in the supplementary file of one of the studies that included [21], detailed raw data of HRV parameters (i.e., RMSSD) for 2 weeks around the vaccination, according to vaccine types, dose of vaccination, and age and sex of the participants were reported. However, the postvaccination change in RMSSD was graphically represented only, which precluded a comparative analysis of these data with those of the other studies.”

(Please refer red words in page 4)

“Besides the meta-analysis, the rate of change of RMSSD from baseline (∆RMSSD (%)) in Presby et al. [21] was calculated and visualized together with the data of Hajduczok et al. [30] on ∆RMSSD (%). Analysis of ∆RMSSD (%) up to the 6th [30] or 7th day [21] after the first and second doses of Pfizer/BioNTech vaccination showed a rapid change within the first 3 days of vaccination in both cases, and a return to the baseline thereafter (Figure 3).”

(Please refer red words in page 9)

2) The rate of change of the RMSSD from baseline (∆RMSSD (%)) in Presby et al. was calculated, and this was visualized together with the data that Hajduczok et al. reported for the ∆RMSSD (%) as their outcome.

(Please refer Figure 3)

3) Additional meta-analysis, regardless of the vaccine type and dose, found that the change in RMSSD values from baseline to that during the 1–3 days after vaccination was particularly heterogeneous.

“Regardless of vaccine type and dose, a meta-analysis of differences of RMSSD value from baseline according to days after vaccination showed high heterogeneity 1–3 days after vaccination (I2 = 85.2–99.4%), which decreased 4 days after vaccination (I2 = 44.1–66.6%) (Figure S1).”

(Please refer red words in page 9)

Comment 6:

  1. (This question need not to be answered)

The purpose of this paper is to “have public health implications in alleviating the hesitancy associated with COVID-19 vaccine uptake by the general population”. Generally speaking, the most concerned topics in this field are allergic reactions and side-effects, as well as the long period safety issue. When focusing on the side-effects, HRV plays a much unimportant role compared with other symptoms, so a lot of related research did not measure it at all, that is why there is so little research data. And for the long period safety, this paper did not talk about it at all. So I can not say the purpose of this paper is well achieved. Still, combining HRV with COVID-19 vaccine safety issues is a very interesting point, the author just need to put more effort in it (narrow down and focus on the main topic, digging the data in a deeper level or another dimension, rephrase some section such as section 2 and 4 to reduce the length of this paper).

Response 6:

Thank you for the comments. We believe that your comment is very incisive and accurate. Thus, we concur that the results of this systematic review should not be interpreted as if they prove the safety of COVID-19 vaccinations. Nonetheless, the changes in HRV may be more relevant to some limited aspects of the safety of COVID-19 vaccines, particularly in the context of postvaccination ANS dysfunctions. Accordingly, we have added the purpose and limitations in the revised manuscript.

“Although it is not an indicator of vaccine safety, HRV may provide insight into [6-11] postvaccination ANS dysfunction. However, no systematic attempt to investigate the relationship of HRV with COVID-19 vaccination or ANS dysfunction has been undertaken.”

(Please refer red words in page 2)

Finally, since this review focused on changes of HRV after COVID-19 vaccination, the results do not guarantee the safety of the COVID-19 vaccine, but only suggest some safety aspects related to COVID-19 vaccination. Nevertheless, the findings of the association between COVID-19 vaccination and HRV provide an insight into the context of ANS dysfunction after COVID-19 vaccination [6-8,11,34].”

(Please refer red words in page 13)

Round 2

Reviewer 2 Report

I have thoroughly read the revised version of this manuscript and all the answers from the authors to my comments. I think all the questions are answered adequately and the manuscript are modified meanwhile.